# Evaluation of usefulness in surfactant protein D as a predictor of mortality in myositis-associated interstitial lung disease

Shinjiro Kaieda[1], Takahisa Gono[2]*, Kenichi Masui[3], Naoshi Nishina[4], Shinji Sato[5], Masataka Kuwana[2], A Multicenter Retrospective Cohort of Japanese Patients with Myositis-associated ILD (JAMI) investigators[¶]

1 Department of Medicine, Division of Respirology, Neurology, and Rheumatology, Kurume University School of Medicine, Fukuoka, Japan, 2 Department of Allergy and Rheumatology, Nippon Medical School Graduate School of Medicine, Tokyo, Japan, 3 Department of Anaesthesiology, Showa University School of Medicine, Tokyo, Japan, 4 Division of Rheumatology, Department of Internal Medicine, Keio University School of Medicine, Tokyo, Japan, 5 Division of Rheumatology, Department of Internal Medicine, Tokai University School of Medicine, Tokyo, Kanagawa, Japan

¶ Membership of JAMI cohort study is provided in the Acknowledgments.
* t-gono@nms.ac.jp

**Data Availability Statement:** Data cannot be shared publicly because of confidential data. Data are available from the Nippon Medical School Institutional Ethics Committee (contact the

## Abstract

### Objective

Surfactant protein D (SP-D) is considered a serum biomarker of various forms of interstitial lung disease (ILD). In this study, we examined the utility of SP-D as a predictive biomarker for mortality in patients with ILD associated with polymyositis/dermatomyositis (PM/DM) using large-scale multicentre cohort data.

### Methods

We enrolled 381 patients with incident PM/DM-associated ILD in a multicentre retrospective cohort based on the availability of serum SP-D at the baseline. Demographic and clinical characteristics as well as the presence of autoantibodies to melanoma differentiation-associated gene 5 (MDA5) and aminoacyl tRNA synthetase were measured at the time of diagnosis, and follow-up survival data were collected prospectively.

### Results

Seventy-eight patients died during the median observation period of 18 months, and the majority of patients died of ILD. The SP-D levels at baseline were significantly lower ($P = 0.02$) in a non-survivor subset than in a survivor subset among the entire enrolled patients. However, the SP-D levels were higher in the non-survivor subset than in the survivor subset based on the stratification by anti-MDA5-positive, anti-ARS-positive and, double-negativity, although there was an only statistically significant difference ($P = 0.01$) in the double-negative group. Surprisingly, the SP-D levels were within the upper limit of normal, 110 ng/mL, in 54 (87%) of 62 anti-MDA5-positive patients who died. In the double-negative group, the mortality rates were significantly higher ($P = 0.002$) in a subset with SP-D $\geq$127.6 ng/mL,

committee by email:nms_fuzokurinri@nms.ac.jp) for researchers who meet the criteria for access to confidential data.

**Funding:** This work was supported by a research grant for intractable diseases from the Japanese Ministry of Health, Labour and Welfare. The funders had no role in the study design, data collection and analysis, decision to publish, or preparation of the manuscript.

**Competing interests:** T.G. received speaker fees from Astellas and the MBL. S.S. holds a patent for the anti-MDA5 antibody measurement kit (publication of JPWO2010024089A1). M.K. holds a patent for the anti-MDA5 antibody measurement kit (publication of JPWO2010024089A1), received research grants from Astellas, and speaker fees from Astellas, the Japan Blood Products Organization, and the MBL. M.K. currently serves as an Academic Editor of PLOS ONE. This does not alter our adherence to PLOS ONE policies on sharing data and materials. The other authors have no conflicts of interest.

the cut-off value for mortality calculated by the receiver operating characteristic curve, than the other subset. All of patients with SP-D <127.6 ng/mL survived.

## Conclusion

Serum SP-D levels behave differently among patients with stratified by anti-MDA5 antibody, anti-ARS antibody and both negativity in PM/DM-associated ILD. Its use in clinical practice should be applied with caution on the basis of the presence or absence of anti-MDA5 antibody or anti-ARS antibody.

## Introduction

Polymyositis and dermatomyositis (PM/DM) are idiopathic inflammatory myopathies characterized by muscle weakness and skin rash, such as Gottron's papules or signs and heliotrope rash [1]. Of the extra-muscular manifestations of PM/DM, interstitial lung disease (ILD) is the leading cause of death [2]. A number of circulating biomarkers have been shown to be useful in assessing disease activity and/or predicting the outcomes of ILD in patients with connective tissue diseases, including PM/DM; these biomarkers include autoantibodies, pneumoproteins, such as Krebs von den Lungen-6 (KL-6) and surfactant protein D (SP-D), and inflammation-related proteins, such as C-reactive protein (CRP) and a variety of cytokines and chemokines [3]. Myositis-specific autoantibodies (MSAs) are the most powerful biomarkers for predicting the clinical presentation, response to treatment, and prognosis in patients with PM/DM [4]. It has been shown that the measurement of additional biomarkers potentially enhances the predictive performance of MSAs. For example, in patients positive for anti-melanoma differentiation-associated gene 5 (MDA5) antibody, mortality due to ILD was higher in patients with hyperferritinaemia than in those without hyperferritinaemia [5]. Therefore, the combined evaluation of multiple biomarkers is used in clinical practice for patients with PM/DM-associated ILD.

We recently established a multicentre retrospective cohort of Japanese patients with PM/DM-associated ILD (JAMI), which involved 44 institutions across Japan, and we successfully identified independent predictors of short-term ILD-related mortality [6]. In this cohort, serum SP-D was identified as one of the predictors of mortality; a high mortality rate was associated with a low level of SP-D. This is inconsistent with the results of previous studies showing negative correlations between SP-D and lung function parameters, such as vital capacity and diffusing capacity for carbon monoxide, in patients with PM/DM-associated ILD [7]. Another study suggested that an increase in the levels of SP-D during the first 4 weeks of immunosuppressive therapy was a risk factor for death in patients with PM/DM-associated ILD [8]. In addition, serum SP-D levels are not associated with malignancy and infectious pneumonia in a way that is different from that of KL-6. Serum SP-D is clinically one of the useful biomarkers related to ILD [9]. In this study, we assessed usefulness of serum SP-D levels on prediction of mortality in patients with PM/DM-associated ILD using the JAMI cohort data, with consideration of the heterogeneity of the disease.

## Patients and methods

### Patients

This study utilized data from the JAMI cohort, which was described in detail elsewhere [6]. Briefly, the JAMI cohort was a nationwide, multicentre retrospective and prospective cohort that consisted of 499 adult patients with PM, classic DM, or clinically amyopathic DM

(CADM) complicated with ILD. We selected 381 patients based on the availability of serum SP-D data at the baseline. The study was approved by the Ethics Committee of the coordinating centre (Nippon Medical School, Tokyo, Japan; 26-03-434) and by individual participating centres. The JAMI cohort had been registered in the University Hospitals Medical Information Network Clinical Trial Registry (UMIN000018663).

### Detection of MSAs

MSA identification was performed centrally in experienced laboratories. Anti-aminoacyl-tRNA synthetase (ARS) antibodies were detected with an RNA immunoprecipitation assay, as described previously [10]. Anti-MDA5 antibody was measured with an in-house enzyme-linked immunosorbent assay using recombinant MDA5 as an antigen source [11].

### Statistical analysis

A P-value $< 0.05$ was regarded as significant. Mann-Whitney *U* test was used to compare median values for continuous data. Survival curves were obtained with the Kaplan-Meier method, and differences in overall survival between subgroups were analysed using the log-rank test. The optimal cut-off values of SP-D for distinguishing survivors from non-survivors were defined on the basis of the receiver-operating characteristic (ROC) curve. All statistical analyses were performed with JMP software (SAS Institute, NC, USA).

## Results

### Baseline clinical characteristics, regimens used for induction treatment and outcomes

The baseline characteristics of the 381 patients with incident PM/DM-associated ILD are shown in Table 1. The mean age (standard deviation) at onset was 56 [12] years old. The number of female patients was 253 (66%). The disease duration at diagnosis was 2 months, indicating that the majority of patients were diagnosed and treated early. Our cohort consisted mainly of patients with classic DM (31%) and CADM (53%), and only 13% were classified as having PM. The baseline median level (interquartile range [IQR]) of SP-D was 91.1 ng/mL (44.9–175.5). Anti-ARS and anti-MDA5 antibodies were detected in 131 (34%) and 170 (45%) patients, respectively. One patient had both anti-MDA5 and anti-ARS antibodies and was thus excluded, while 81 (21%) were negative for both MSAs ("double-negative"). High-dose corticosteroids (CS) in combination with immunosuppressants were commonly used as regimens for induction treatment and included the "double combo" with cyclophosphamide (CYC) in 13 (3%) patients, the "double combo" with a calcineurin inhibitor in 146 (38%) patients, and the "triple combo" with CYC and a calcineurin inhibitor in 166 (44%) patients.

In total, 78 patients, including 62 with anti-MDA5, 8 with anti-ARS, and 8 with double-negative, died during the median observation period of 18 months. The overall survival rates were 81% and 75% at 1 and 3 years, respectively. The causes of death included respiratory insufficiency directly related to ILD in 68 (87%) patients, infection in 2 (3%) patients, malignancy in 3 (4%) patients, and other causes in 5 (6%) patients, indicating that the majority of deaths were the result of ILD in our patient population.

### Comparison of the SP-D level at baseline between survivors and non-survivors

The SP-D levels at baseline were significantly lower ($P = 0.02$) in the non-survivor subset; median level (IQR), 72.5 (34.4–133.3), than in the survivor subset; median level (IQR), 99.5

**Table 1. Baseline characteristics and regimens used for induction treatment in 381 patients with PM/DM-associated ILD.**

| Variables | Value | Available data per outcome |
|---|---|---|
| Demographics | | |
| Age at onset, years | 56 ± 12 | 381 (100%) |
| Female, no (%) | 253 (66) | 381 (100%) |
| Disease duration at diagnosis, month | 2 [1–5] | 381 (100%) |
| Diagnosis | | |
| PM, no (%) | 48 (13) | 381 (100%) |
| Classic DM, no (%) | 119 (31) | 381 (100%) |
| CADM, no (%) | 214 (53) | 381 (100%) |
| Pulmonary function testing | | |
| %Vital capacity, predicted | 77 (63–92) | 309 (81%) |
| %DLco, predicted | 61 (48–78) | 262 (69%) |
| Laboratory parameters | | |
| CRP, mg/dL | 0.7 (0.2–1.8) | 378 (99%) |
| Ferritin, ng/mL | 348 (134–721) | 285 (75%) |
| KL-6, U/mL | 830 (547–1296) | 381 (100%) |
| SP-D, ng/mL | 91 (45–175) | 381 (100%) |
| MSAs | | |
| Anti-ARS antibody, no (%) | 131 (34)* | 381 (100%) |
| Anti-MDA5 antibody, no (%) | 170 (45)* | 381 (100%) |
| Double-negative, no (%) | 81 (21) | 381 (100%) |
| Regimens used for induction treatment | | |
| CS alone, no (%) | 56 (15%) | 381 (100%) |
| CS + IVCY, no (%) | 13 (3%) | 381 (100%) |
| CS + CNI, no (%) | 146 (38%) | 381 (100%) |
| CS + IVCY + CNI, no (%) | 166 (44%) | 381 (100%) |

Continuous variables are shown as median [interquartile range].

*One patient was positive for both anti-ARS and MDA5 antibodies.

PM: polymyositis, DM: dermatomyositis, CADM: clinically amyopathic dermatomyositis, DLco: diffusing capacity for carbon monoxide, CRP: C-reactive protein, MSA: myositis-specific autoantibody, ARS: aminoacyl tRNA synthetase, MDA5: melanoma differentiation-associated gene 5, CS: corticosteroid, IVCY: intravenous cyclophosphamide, CNI: calcineurin inhibitor

(46.7–189.9) (Fig 1A). In the non-survivor subset, there was no statistically significant difference (P = 0.17) in the SP-D levels between patients who died due to ILD and those who died due to the other causes. Thus, SP-D levels were not influenced by the other factors such as malignancy and infectious pneumonia, suggesting this result was similar to those in previous reports.

## Comparison of the SP-D level, disease duration and pulmonary function at baseline among MSAs

In our cohort, 78% of patients who died had anti-MDA5 antibody. We analysed the baseline SP-D levels based on the stratification by MSAs. The baseline median level (IQR) of serum SP-D was 51 (29.5–77.0), 169.2 (1–4.9–260.4), and 124 (52.2–242.5) in anti-MDA5 antibody-positive group, anti-ARS-antibody positive group and, double-negative group, respectively (Fig 2A). The SP-D levels were significantly lower in anti-MDA5 antibody-positive group than

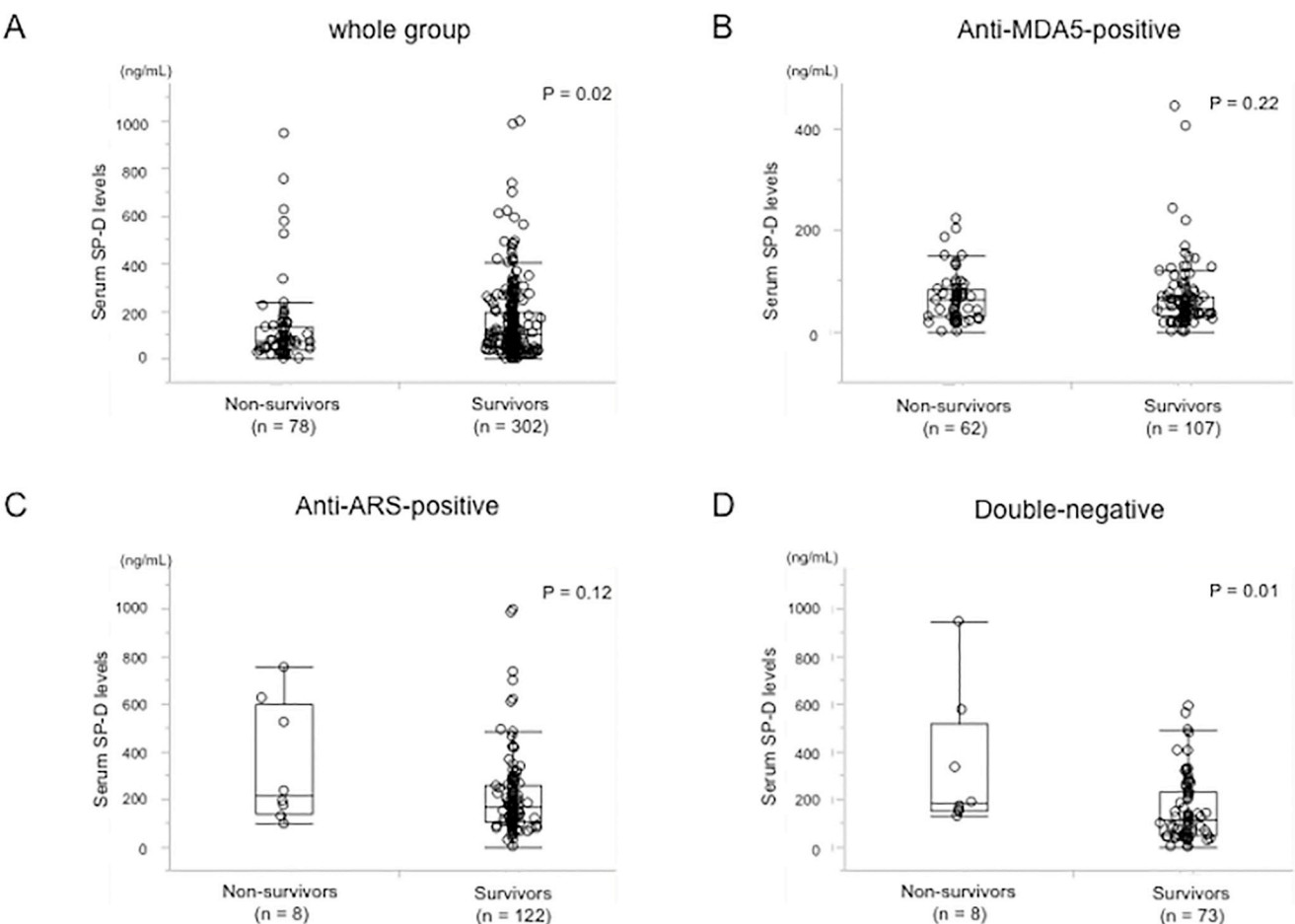

**Fig 1. Comparison of the SP-D level at baseline between survivors and non-survivors in various patient populations with PM/DM-associated ILD.**
The median baseline levels of SP-D were compared between survivors and non-survivors in the whole cohort (A), patients with anti-MDA-5 antibody (B), patients with anti-ARS antibody (C), and patients with double-negative (D). MDA5: melanoma differentiation-associated gene 5, ARS: aminoacyl tRNA synthetase.

in anti-ARS antibody positive group ($P < 0.0001$) or double-negative group ($P < 0.0001$). Surprisingly, the SP-D levels increased over the upper limit of normal, 110 ng/mL, just only in 24 (14%) of 169 anti-MDA5 patients. On the other hand, the increased SP-D levels with more than 110 ng/mL were revealed in 96 (74%) of 130 anti-ARS patients and 45 (56%) of 81 double negative patients, respectively. The SP-D levels were significantly highest in anti-ARS antibody-positive group among these 3 groups (Fig 2A).

We also compared disease duration and pulmonary function among the three groups. With regard to disease duration at diagnosis of PM/DM-associated ILD, the median disease duration (IQR) was significantly shorter in anti-MDA5 antibody-positive group: 2 months [1–3], than in anti-ARS antibody-positive group: 3 months (1–8) ($P = 0.0002$) or double-negative group: 3 months [1–6] ($P = 0.0005$). The range of disease duration in anti-ARS antibody-positive group was 0–122 months; longer than that of double-negative group with 0–88 months, although there was no significant difference of the median disease duration ($P = 0.82$) between the two groups. In terms of the severity of pulmonary dysfunction with %vital capacity (VC), the median value (IQR) of %VC was lower in anti-ARS antibody positive group: 70.2% (60–85), than in double-negative group: 84.9% (66–98) ($P = 0.0002$) or anti-MDA5 antibody

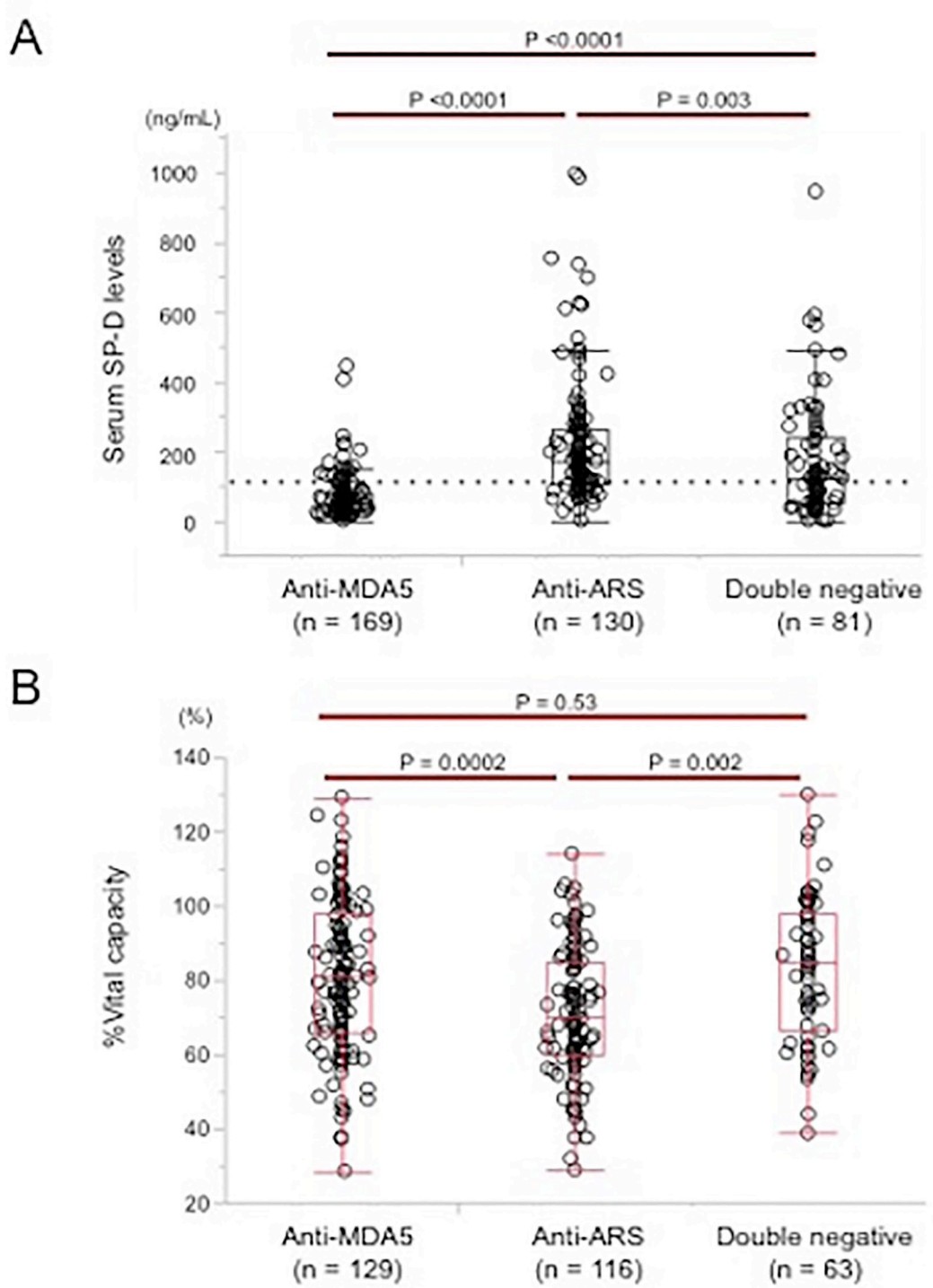

**Fig 2. Comparison of the SP-D level and %vital capacity at baseline among MSAs.** The median baseline levels of SP-D
(A) and %vital capacity (B) were compared between patients with anti-MDA-5 antibody, patients with anti-ARS antibody,
and patients with double-negativity. Broken lines in Fig 2A indicate the upper limit of normal in serum SP-D levels, which
was 110 ng/mL. MDA5: melanoma differentiation-associated gene 5, ARS: aminoacyl tRNA synthetase.

positive group: 81% (66–98) (*P* = 0.0005) (Fig 2B). On the other hand, there was no significant difference (*P* = 0.53) between anti-MDA5 antibody-positive group and double-negative group. These results indicated more widespread damage of ILD as well as longer disease duration in anti-ARS antibody-positive group than the other two groups.

## Comparison of the SP-D level at baseline between survivors and non-survivors based on stratification by MSA

Based on the stratification by MSA, the SP-D levels were higher in the non-survivor subset than in the survivor subset in each MSA group: anti-MDA5-positive group (Fig 1B), anti-ARS-positive group (Fig 1C) and, double-negative group, respectively (Fig 1D), although there was an only statistically significant difference in double-negative group (*P* = 0.01). Among the entire 78 non-survivors, the increased SP-D levels with more than 110 ng/mL were revealed in 8 (13%) of 62 anti-MDA5-positive patients, 7 (88%) of 8 anti-ARS-positive patients, and 8 (100%) of 8 double–negative patients, respectively. In double-negative patients, the ROC curve analysis identified that the optimal cut-off value of the SP-D levels for predicting all-cause mortality was 127.6 ng/mL (AUC 0.77, *P* = 0.002). Among double-negative patients, 8 (20%) of 38 patients with SP-D ≥127.6 ng/mL died (Fig 3). On the other hand, all of the 43 patients with SP-D <127.6 ng/mL survived.

## Discussion

This study identified for the first time that serum SP-D levels behaved differently among patients with PM/DM-associated ILD stratified by anti-MDA5 antibody, anti-ARS antibody

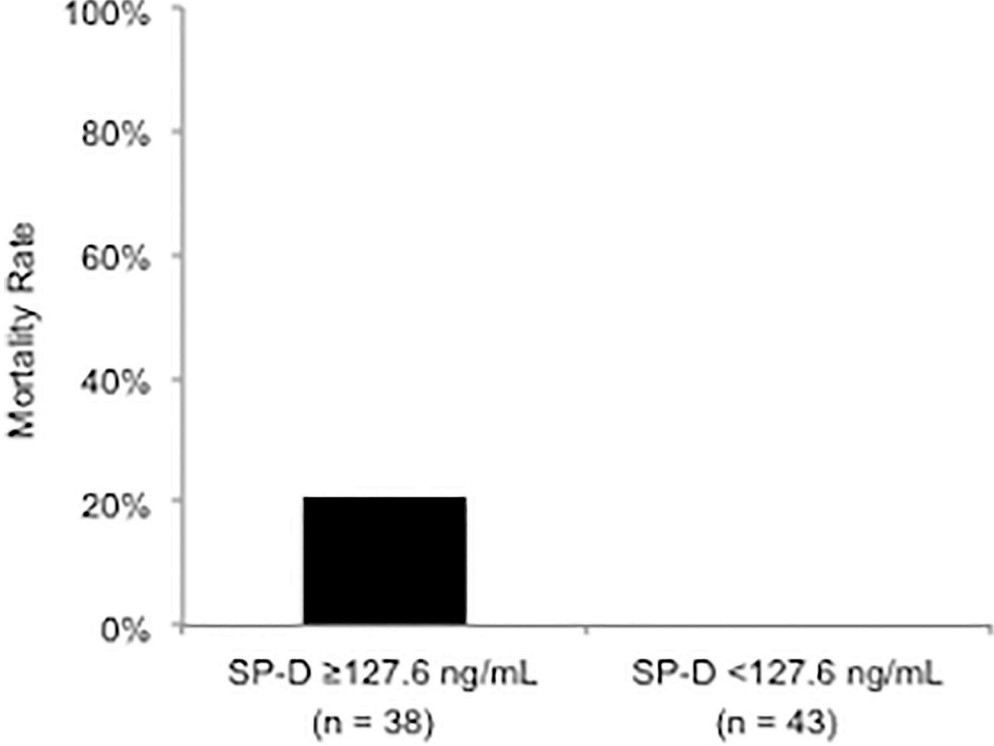

**Fig 3. Mortality rate in patients with ≥127.6 ng/mL of the SP-D level and those without in double-negative group.** 127.6 ng/mL at the baseline serum level of SP-D was calculated as the cut-off value for prediction of all-cause mortality by the ROC curve analysis among double-negative patients.

and both negativity. In accordance with previous reports [7, 8], this study demonstrated that the serum level of SP-D at baseline were higher in patients with poor prognosis than those without (Fig 1B–1D). However, in the whole group, serum SP-D levels were lower in the non-survivors than the survivors. Our previous study has demonstrated that an SP-D level <95.4 ng/mL was identified as a predictor of mortality due to ILD in the analysis of the entire patient population enrolled in the JAMI cohort [6]. In JAMI cohort, anti-MDA5 antibody was positive in approximately 80% of non-survivors. This study also showed that the serum SP-D levels were lowest in anti-MDA5-positive group among groups regardless of survivors or non-survivors (Fig 2). In addition, surprisingly, the SP-D levels rose over the upper limit of normal just only in 13% of anti-MDA5-positive patients who died. Thus, the skewed distribution of non-survivors with anti-MDA5 antibody influenced on the lower levels of serum SP-D in the entire non-survivors in JAMI cohort.

This study has elucidated that the SP-D levels were within the upper limit of normal in approximately 90% of anti-MDA5 patients. The mechanisms underlying serum lower SP-D levels in patients with anti-MDA5 antibody remain unknown. SP-D is a surfactant protein secreted mainly from type II pneumocytes and is involved in stabilizing alveolar surface tension at the air-liquid interface and supporting lung host innate immunity through the direct killing of bacteria and enhancement of phagocytosis [12]. The elevated serum levels of SP-D in patients with various forms of ILD could be attributed to an increase in SP-D production by regenerating alveolar type II pneumocytes and/or to the enhanced permeability of the air-blood barrier following the destruction of alveolar capillaries in the affected lung [13]. Histopathological differences of ILD at diagnosis of PM/DM-associated ILD might contribute to the behavior of serum SP-D levels in each MSA group. In this study, the disease duration was significantly longer in in anti-ARS antibody-positive group or double-negative group than in anti-MDA5-positive group. In addition, the median value of %vital capacity was lower in anti-ARS antibody-positive group than in double-negative group or anti-MDA5 antibody-positive group. These findings might indicate that chronic damage and remodelling of peripheral pulmonary structure was more apparent in anti-ARS antibody-positive group, causing translocation of SP-D molecules from alveolar epithelium to blood vessels much more. On the other hand, the intrinsic mechanism regarding lower SP-D levels is potentially relevant to the rapid progressive and fatal course of anti-MDA5 antibody-associated ILD. This might be explained by impaired regeneration of type II pneumocytes and/or impaired production of SP-D by regenerating type II pneumocytes, as reported in patients with idiopathic pulmonary fibrosis [14].

In addition, this study demonstrated that the measurement of serum SP-D was useful for the prediction of outcomes in the double-negative group, suggesting that the double-negative patients with lower levels of SP-D have favorable prognosis. SP-D can be measured as a laboratory test for clinical use in Japan based on the unique Japanese health insurance system. The measurement of SP-D is not yet available at clinical practice in many countries. Further study is required to investigate the role of SP-D in the pathogenic process of PM/DM-associated ILD in patients with anti-MDA5 antibody, and to validate whether the measurement of serum SP-D levels is beneficial for prediction of outcomes in each MSA among patients with PM/DM-associated ILD in daily practice if the test is approved for clinical use in many countries over the world.

The present study has several potential limitations. First, because patients were selected mainly from tertiary referral hospitals, selection bias towards a more severe form of the disease, such as anti-MDA5 antibody-associated rapidly progressive ILD, could influence the results. Secondly, we lacked data regarding SP-D in 118 out of 499 patients enrolled in JAMI cohort because this study was retrospectively conducted, and the measurement of SP-D was

made based on decisions by physicians who participated in this study. In addition, the median observation period in the JAMI cohort was 18 months, and the biomarkers identified in this study were primarily short-term prognostic markers.

In conclusion, serum SP-D levels behave differently among patients with stratified by anti-MDA5 antibody, anti-ARS antibody and both negativity in PM/DM-associated ILD. Its use in clinical practice should be applied with caution on the basis of the presence or absence of anti-MDA5 antibody or anti-ARS antibody.

## Acknowledgments

**Other JAMI investigators**: Yasushi Kawaguchi (Tokyo Women's Medical University); Atsushi Kawakami (Nagasaki University Graduate School of Biomedical Sciences); Kei Ikeda (Chiba University Hospital); Maasa Tamura, Yohei Kirino, and Yukie Yamaguchi (Yokohama City University Graduate School of Medicine); Yoshinori Tanino (Fukushima Medical University School of Medicine); Takahiro Nunokawa (Tokyo Metropolitan Tama Medical Center); Yuko Kaneko (Keio University School of Medicine); Katsuaki Asakawa (Niigata University Medical and Dental Hospital); Taro Ukichi (The Jikei University School of Medicine); Taio Naniwa (Nagoya City University School of Medicine); Yutaka Okano (Kawasaki Municipal Kawasaki Hospital); Yoshinori Taniguchi (Kochi Medical School, Kochi University); Jun Kikuchi (Saitama Medical Center, Saitama Medical University); Makoto Kubo (Yamaguchi University Graduate School of Medicine); Masaki Watanabe (Graduate School of Medical and Dental Sciences, Kagoshima University); Tatsuhiko Harada (Nagasaki University School of Medicine); Taisuke Kazuyori (The Jikei University School of Medicine Katsushika Medical Center); Hideto Kameda (Toho University Ohashi Medical Center); Makoto Kaburaki (Toho University School of Medicine); Yasuo Matsuzawa (Toho University Medical Center, Sakura Hospital); Shunji Yoshida (Fujita Health University School of Medicine); Yasuko Yoshioka (Juntendo University Urayasu Hospital); Takuya Hirai (Juntendo University Urayasu Hospital); Yoko Wada (Niigata University Graduate School of Medical and Dental Sciences); Koji Ishii (Faculty of Medicine, Oita University); Sakuhei Fujiwara (Faculty of Medicine Oita University); Takeshi Saraya (Kyorin University); Kozo Morimoto (Fukujuji Hospital, Japan Anti-Tuberculosis Association); Tetsu Hara (Hiratsuka Kyosai Hospital); Hiroki Suzuki (Saiseikai Yamagata Saisei Hospital); Hideki Shibuya (Tokyo Teishin Hospital); Yoshinao Muro (Nagoya University Graduate School of Medicine); Ryoichi Aki (Kitasato University School of Medicine); Takuo Shibayama (National Hospital Organization Okayama Medical Center); Shiro Ohshima (National Hospital Organization Osaka Minami Medical Center); Yuko Yasuda (Saiseikai Kumamoto Hospital); Masaki Terada (Saiseikai Niigata Daini Hospital); and Yoshie Kawahara (Keiyu Hospital).

## Author Contributions

**Conceptualization:** Shinjiro Kaieda, Takahisa Gono, Kenichi Masui, Naoshi Nishina, Shinji Sato, Masataka Kuwana.

**Data curation:** Shinjiro Kaieda, Takahisa Gono, Kenichi Masui, Naoshi Nishina, Shinji Sato.

**Formal analysis:** Takahisa Gono, Kenichi Masui.

**Investigation:** Shinjiro Kaieda, Takahisa Gono, Kenichi Masui, Naoshi Nishina, Shinji Sato, Masataka Kuwana.

**Project administration:** Masataka Kuwana.

**Writing – original draft:** Shinjiro Kaieda, Takahisa Gono.

**Writing – review & editing:** Takahisa Gono, Kenichi Masui, Naoshi Nishina, Shinji Sato, Masataka Kuwana.

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
