## [Decision Letter · Decision Letter 0]

21 Apr 2020

PONE-D-20-07783

Evaluation of usefulness in surfactant protein D as a predictor of mortality in myositis-associated interstitial lung disease

PLOS ONE

Dear Dr. Gono,

Thank you for submitting your manuscript to PLOS ONE. After careful consideration, we feel that it has merit but does not fully meet PLOS ONE’s publication criteria as it currently stands. Therefore, we invite you to submit a revised version of the manuscript that addresses the points raised during the review process.

We would appreciate receiving your revised manuscript by Jun 05 2020 11:59PM. To enhance the reproducibility of your results, we recommend that if applicable you deposit your laboratory protocols in protocols.io, where a protocol can be assigned its own identifier (DOI) such that it can be cited independently in the future. For instructions see: http://journals.plos.org/plosone/s/submission-guidelines#loc-laboratory-protocols

We look forward to receiving your revised manuscript.

Kind regards,

Minghua Wu, M.D., Ph.D.

Academic Editor

PLOS ONE

Journal Requirements:

2. Thank you for including the following consent information on the submission details page:

'Consent was not obtained because the data were analyzed anonymously.'

Please also include this information in the ethics statement in the Methods section of your manuscript.

4. We note that you have a patent relating to material pertinent to this article.

a. Please provide an amended statement of Competing Interests to declare this patent (with details including name and number), along with any other relevant declarations relating to employment, consultancy, patents, products in development or modified products etc.

Please confirm that this does not alter your adherence to all PLOS ONE policies on sharing data and materials, as detailed online in our guide for authors http://journals.plos.org/plosone/s/competing-interests by including the following statement: "This does not alter our adherence to  PLOS ONE policies on sharing data and materials.” If there are restrictions on sharing of data and/or materials, please state these.

Please note that we cannot proceed with consideration of your article until this information has been declared.

Reviewers' comments:

Reviewer's Responses to Questions

**Comments to the Author**

1. Is the manuscript technically sound, and do the data support the conclusions?

Reviewer #1: Yes

Reviewer #2: Yes

2. Has the statistical analysis been performed appropriately and rigorously? 

Reviewer #1: Yes

Reviewer #2: Yes

3. Have the authors made all data underlying the findings in their manuscript fully available?

Reviewer #1: Yes

Reviewer #2: Yes

4. Is the manuscript presented in an intelligible fashion and written in standard English?

Reviewer #1: Yes

Reviewer #2: Yes

5. Review Comments to the Author

Reviewer #1: Comments for Authors

Kaieda S et al. examined the utility of SP-D as a predictive biomarker for mortality in patients with ILD associated with PM/DM using JAMI cohort. Overall the manuscript is on important topic and is well written and analyzed with clarity and conviction. I have an only minor concern which authors need to answer.

Minor

1. I agree with authors lower SP-D levels is potentially relevant to the rapid progressive course of anti-MDA5 antibody-associated ILD and the level could be low at the disease diagnosis. However, is there any speculation on the issue which patients with anti-ARS had a higher level of SP-D among other MSAs? Since the elevated serum levels of SP-D could be attributed to an increase in SP-D production by regenerating alveolar type II pneumocytes, did the patients with anti-ARS have already widespread findings of ILD on chest CT at diagnosis or have longer disease duration from diagnosis of ILD?

Reviewer #2: Kaieda S, et al. investigated serum SP-D as a predictive biomarker for mortality in patients with PM/DM-associated ILDs. They also performed stratified analysis by anti-MDA5, anti-ARS, or double-negative and identified a cut-off value at 127.6 ng/ml for their mortality in double-negative group. Indeed, SP-D was not significantly predictive in patients with anti-MDA5 and anti-ARS. This is nicely designed and well performed clinical study that inform usefulness of serum SP-D in PM/DM-associated ILDs. However, several points to strengthen then manuscript were as below:

#1 Please provide intra-class comparison of non-survivor vs. survivor in each MSA group.

#2 It should be emphasized that serum SP-D is approved as a laboratory test for clinical use and easily available at clinical practice in Japan, however it is approved only for research use in many countries other than Japan including Europe and North America. Also, it should be pronounced that serum SP-D might be beneficial for patients outside Japan if the test were approved for clinical use.

#3 Please enrich discussion about reasons for the difference in serum SP-D levels by MSA. Especially, focusing on mechanisms of translocation of SP-D molecules from alveolar epithelium to serum and on difference in damage of peripheral pulmonary structure by MSAs would be preferable.

6. PLOS authors have the option to publish the peer review history of their article (what does this mean?). If published, this will include your full peer review and any attached files.

Reviewer #1: No

Reviewer #2: No

---

## [Author Response · Author response to Decision Letter 0]

15 May 2020

Dear Editor and Reviewers

Thank you for giving us your thoughtful comments/suggestions. We have responded to them one by one, and revised our manuscript according to the manner as below. We are looking forward to your positive consideration of the publication in our manuscript.

Best regards,

Takahisa Gono, corresponding author

Reviewer #1: 

1. I agree with authors lower SP-D levels is potentially relevant to the rapid progressive course of anti-MDA5 antibody-associated ILD and the level could be low at the disease diagnosis. However, is there any speculation on the issue which patients with anti-ARS had a higher level of SP-D among other MSAs? Since the elevated serum levels of SP-D could be attributed to an increase in SP-D production by regenerating alveolar type II pneumocytes, did the patients with anti-ARS have already widespread findings of ILD on chest CT at diagnosis or have longer disease duration from diagnosis of ILD?

Response: I appreciate your valuable comments. As the reviewer suggested, the elevated serum levels of SP-D could be dependent on the degree of regenerating alveolar type II pneumocytes and/or that of destruction in alveolar capillaries. To answer the reviewer’s suggestions, we compared disease duration at diagnosis of ILD among three groups: anti-MDA5 antibody-positive group, anti-ARS antibody-positive group and, double-negative group. The median disease duration (interquartile range [IQR]) was 2 months (1-3), 3 months (1-8), and 3 months (1-6) in anti-MDA5 antibody-positive group, anti-ARS antibody-positive group and, double-negative group, respectively. The disease duration was significantly shorter in anti-MDA5 antibody-positive group than in anti-ARS antibody-positive group (P = 0.0002) or double-negative group (P = 0.0005). The range of disease duration in anti-ARS antibody-positive group was 0-122 months; longer than that of double-negative group with 0-88 months, although there was no significant difference (P = 0.82) between the two groups. 

In terms of the severity of pulmonary dysfunction with %vital capacity (VC), the median value (IQR) of %VC was lower in anti-ARS antibody-positive group: 70.2% (60-85), than in double-negative group: 84.9% (66-98) (P = 0.0002) or anti-MDA5 antibody positive group: 81% (66-98) (P = 0.0005) (revised Figure 2B). On the other hand, there was no significant difference (P = 0.53) between double-negative group and anti-MDA5 antibody-positive group. Unfortunately, we have no data regarding chest HRCT score.

 These findings indicated that the serum levels of SP-D in anti-ARS antibody-positive group were influenced by more widespread damage of ILD as well as the longer disease duration of ILD.

To describe this point, we have added the following sentences as below. 

In line 168 to 182 on page 7 to 8,

“We also compared disease duration and pulmonary function among the three groups. With regard to disease duration at diagnosis of PM/DM-associated ILD, the median disease duration (IQR) was significantly shorter in anti-MDA5 antibody-positive group: 2 months (1-3), than in anti-ARS antibody-positive group: 3 months (1-8) (P = 0.0002) or double-negative group: 3 months (1-6) (P = 0.0005). The range of disease duration in anti-ARS antibody-positive group was 0-122 months; longer than that of double-negative group with 0-88 months, although there was no significant difference of the median disease duration (P = 0.82) between the two groups. In terms of the severity of pulmonary dysfunction with %vital capacity (VC), the median value (IQR) of %VC was lower in anti-ARS antibody positive group: 70.2% (60-85), than in double-negative group: 84.9% (66-98) (P = 0.0002) or anti-MDA5 antibody positive group: 81% (66-98) (P = 0.0005) (Figure 2B). On the other hand, there was no significant difference (P = 0.53) between anti-MDA5 antibody-positive group and double-negative group. These results indicated more widespread damage of ILD as well as longer disease duration in anti-ARS antibody-positive group than the other two groups.”

In line 222 to 229 on page 9,

 “Histopathological differences of ILD at diagnosis of PM/DM-associated ILD might contribute to the behavior of serum SP-D levels in each MSA group. In this study, the disease duration was significantly longer in in anti-ARS antibody-positive group or double-negative group than in anti-MDA5-positive group. In addition, the median value of %vital capacity was lower in anti-ARS antibody-positive group than in double-negative group or anti-MDA5 antibody-positive group. These findings might indicate that chronic damage and remodelling of peripheral pulmonary structure was more apparent in anti-ARS antibody-positive group, causing translocation of SP-D molecules from alveolar epithelium to blood vessels much more.”

Reviewer #2: 

1. Please provide intra-class comparison of non-survivor vs. survivor in each MSA group.

Response: Please see Figure 1B, 1C and 1D. We have provided the intra-class comparison of serum SP-D levels between non-survivors and survivors in each MSA group.

#2 It should be emphasized that serum SP-D is approved as a laboratory test for clinical use and easily available at clinical practice in Japan, however it is approved only for research use in many countries other than Japan including Europe and North America. Also, it should be pronounced that serum SP-D might be beneficial for patients outside Japan if the test were approved for clinical use.

Response: As the reviewer indicated, actually, the measurement of SP-D is not yet available at clinical practice in many countries. In the future, however, it should be validated whether the measurement of serum SP-D levels is beneficial for prediction of outcomes in patients with PM/DM-associated ILD if the test were approved for clinical use in many countries over the world.

To emphasize this issue, we added the following sentences as below.

In line 237 to 244 on page 10, 

“SP-D can be measured as a laboratory test for clinical use in Japan based on the unique Japanese health insurance system. The measurement of SP-D is not yet available at clinical practice in many countries. Further study is required to investigate…, and to validate whether the measurement of serum SP-D levels is beneficial for prediction of outcomes in each MSA among patients with PM/DM-associated ILD in daily practice if the test is approved for clinical use in many countries over the world.”

#3 Please enrich discussion about reasons for the difference in serum SP-D levels by MSA. Especially, focusing on mechanisms of translocation of SP-D molecules from alveolar epithelium to serum and on difference in damage of peripheral pulmonary structure by MSAs would be preferable.

Response: Thank you for your constructive suggestion. Histopathological analysis was not conducted in our study. As the reviewer indicated, we speculate that the histopathological difference of ILD at diagnosis may contribute to the behavior of serum SP-D levels in each MSA. Furthermore, we analyzed disease duration and severity of pulmonary dysfunction with %vital capacity among the three groups, as the reviewer#1 suggested. The disease duration at diagnosis is significantly shorter in anti MDA-5 antibody-positive group than in anti-ARS antibody-positive group or double negative group. In terms of %vital capacity, the median value (interquartile range [IQR]) of %vital capacity was lower in anti-ARS antibody positive group: 70.2% (60-85), than in double-negative group: 84.9% (66-98) (P = 0.0002) or anti-MDA5 antibody positive group: 81% (66-98) (P = 0.0005) (revised Figure 2B). On the other hand, there was no significant difference (P = 0.53) between double-negative group and anti-MDA5 antibody-positive group. These findings might indicate that chronic damage and remodelling of peripheral pulmonary structure was more apparent in anti-ARS antibody-positive group, causing translocation of SP-D molecules from alveolar epithelium to serum much more.

To describe this point, we have added the following sentences as below.

In line 168 to 182 on page 7 to 8,

“We also compared disease duration and pulmonary function among the three groups. With regard to disease duration at diagnosis of PM/DM-associated ILD, the median disease duration (IQR) was significantly shorter in anti-MDA5 antibody-positive group: 2 months (1-3), than in anti-ARS antibody-positive group: 3 months (1-8) (P = 0.0002) or double-negative group: 3 months (1-6) (P = 0.0005). The range of disease duration in anti-ARS antibody-positive group was 0-122 months; longer than that of double-negative group with 0-88 months, although there was no significant difference of the median disease duration (P = 0.82) between the two groups. In terms of the severity of pulmonary dysfunction with %vital capacity (VC), the median value (IQR) of %VC was lower in anti-ARS antibody positive group: 70.2% (60-85), than in double-negative group: 84.9% (66-98) (P = 0.0002) or anti-MDA5 antibody positive group: 81% (66-98) (P = 0.0005) (Figure 2B). On the other hand, there was no significant difference (P = 0.53) between anti-MDA5 antibody-positive group and double-negative group. These results indicated more widespread damage of ILD as well as longer disease duration in anti-ARS antibody-positive group than the other two groups.”

In line 222 to 229 on page 9,

 “Histopathological differences of ILD at diagnosis of PM/DM-associated ILD might contribute to the behavior of serum SP-D levels in each MSA group. In this study, the disease duration was significantly longer in in anti-ARS antibody-positive group or double-negative group than in anti-MDA5-positive group. In addition, the median value of %vital capacity was lower in anti-ARS antibody-positive group than in double-negative group or anti-MDA5 antibody-positive group. These findings might indicate that chronic damage and remodelling of peripheral pulmonary structure was more apparent in anti-ARS antibody-positive group, causing translocation of SP-D molecules from alveolar epithelium to blood vessels much more.”

---

## [Editor Report · Decision Letter 1]

28 May 2020

Evaluation of usefulness in surfactant protein D as a predictor of mortality in myositis-associated interstitial lung disease

PONE-D-20-07783R1

Dear Dr. Gono

We are pleased to inform you that your manuscript has been judged scientifically suitable for publication and will be formally accepted for publication once it complies with all outstanding technical requirements.

With kind regards,

Minghua Wu, M.D., Ph.D.

Academic Editor

PLOS ONE
---

## [Editor Report · Acceptance letter]

1 Jun 2020

PONE-D-20-07783R1 

Evaluation of usefulness in surfactant protein D as a predictor of mortality in myositis-associated interstitial lung disease 

Dear Dr. Gono:

I am pleased to inform you that your manuscript has been deemed suitable for publication in PLOS ONE. Congratulations! Your manuscript is now with our production department. 

With kind regards,

on behalf of

Dr. Minghua Wu 

Academic Editor

PLOS ONE